# Understanding Perceived Site Qualities and Experiences of Urban Public Spaces: A Case Study of Social Media Reviews in Bryant Park, New York City

**Yang Song** [1] , **Jessica Fernandez** [2] **and Tong Wang** [3],*

[1]  Department of Landscape Architecture and Urban Planning, Texas A&M University,
    College Station, TX 77840, USA; yangsong@tamu.edu
[2]  College of Environment + Design, University of Georgia, Athens, GA 30602, USA;
    Jessica.Fernandez@uga.edu
[3]  School of Architecture and Urban Planning, Huazhong University of Science and Technology,
    Wuhan 430074, China
*   Correspondence: wangtong@hust.edu.cn

**Abstract:** Urban public spaces are a key component to the well-being and prosperity of modern society. It has been increasingly important to improve the qualities and maximize the usages of urban public spaces. There is a lack of studies that investigate how people use and perceive urban parks using quantitative analysis of location-based social media reviews. This study tackles this gap by introducing a case study that uses social media reviews (Tripadivisor.com) to understand the perceived site quality and experiences of Bryant Park in New York City. A large dataset including 11,419 Tripadvisor reviews from 10,615 users was collected. LDA (Latent Dirichlet Allocation), a natural language processing and machine learning technique, was used to perform topic modeling analysis that could reveal hidden themes in large amounts of text. The results include five semantic topics and their associated topic terms. A comprehensive overview of the user experiences in Bryant Park were provided along with their weekly and monthly dynamics. The findings provide insights for future public space designers and managers by revealing how users describe the designs and operations of Bryant Park.

**Keywords:** social media; public space; Bryant Park; site qualities; experiences

## 1. Introduction

### 1.1. The Quality and Experiences of Urban Public Spaces

Urban public spaces including parks, greenspaces, and streets can promote physical activities, improve mental health [1], increase social strength [2,3], and bring economic and ecological value to local communities [4,5]. To maximize these benefits, a primary goal for city planners, park managers, and landscape architects is to increase the access, usage, and engagement of public spaces. The Center for Disease Control and Prevention (CDC) and World Health Organization (WHO) released guidelines that open or green space could assist to encourage physical activity [6,7]. The Trust for Public Land released rankings for the evaluation of park access and quality of the 100 largest US cities from 2012–2019 [8]. Many studies also look at the associations between characteristics such as size, proximity to the neighborhood, number of amenities, programming, and safety with park usage and physical activities [9,10].

While these attributes play a big role in the success of urban public spaces, specific qualitative factors such as real-time experiences and perceived qualities might be a stronger predictor than hard

measured parameters [11,12]. Theories of sense of place and environmental perception have established concepts that explain the connections between human perceptions and environments [13,14]. These help many practitioners such as public space promoters, city park managers, landscape architects, and urban designers to create vibrant public outdoor spaces. For example, Gehl Institute (https://gehlpeople.com/) defined 12 quality criteria tools for public lives. Project for Public Spaces (www.pps.org) also developed the Power of 10 theory and its place-making process. Both organizations have created successful urban projects that transformed a wide range of communities.

### 1.2. Learn from the Public

Researchers need to investigate peoples' uses, behaviors, satisfaction, psychological perceptions, and sociability in public spaces to gain a deep understanding of public space qualities and experiences, which may help inform urban practices and evaluate site performances [15]. The local community can be a great resource in this process since public space demands a democratic approach [16]. Empowering the public and reducing the institutional power on public space may provide a more relevant and actionable understanding of public spaces to users [17]. Many studies have focused on this participatory approach. Lynch (1960) [18] initiated the idea of having residents map their cities to highlight the key elements of urban space (i.e., paths, edges, districts, nodes, and landmarks), revealing structured principles of how people recognize their environments. Whyte (1980) [19] started his observation methods through videotaping the parks, squares, and streets in New York City, and his findings on how people interact with each other and the surroundings have a profound influence on landscape architecture and urban design practice today. Recently, Gill, Lange, Morgan, and Romano (2013) [20] introduced new media communication technologies to engage stakeholders and facilitate participatory planning and design processes, which assist the understanding of site qualities and experiences. Schlickman and Domlesky (2019) [21] have used modern photography and machine learning technologies to observe site dynamics and user behaviors in ten recently constructed publicly and privately owned sites in Manhattan. However, it is still difficult to conduct these studies on a large scale since on-site observations, surveys, workshops, and interviews pose challenges including a high cost and time commitment, low response rate, outdated information, oversimplification, and consistency issues [22,23].

### 1.3. Social Media Research in Existing Urban Public Space Research

The convenience of mobile phones and wireless services results in millions of daily active users who post images, videos, reviews, and hashtags in real time on platforms such as Instagram, Twitter, Tripadvisor, and Google Places. Since social media data has a high volume, richness, and tremendous speed [24], there are large opportunities that use humans as sensors for researchers and policy makers to understand public life and opinions [25]. There is also evidence that the variations in social media data are highly correlated with real world patterns, as shown in a recent study comparing the relationships between park visitations and social media postings in Minneapolis, Minnesota [26].

On the other hand, social media data are often 'loose', 'noisy', and 'scattered' since they contain user-generated and self-motivated content that are driven by self-representation [27]. This is different from typical surveys, which often have focused sample groups and ask targeted questions. Therefore, social media data or big data does not necessarily equate to high-quality data, though data analysis and interpretation are still key elements for research purposes [28].

Many urban planning and environmental researchers have recently adopted social media data into their research. Using Twitter, a microblogging platform, Roberts, Sadler, and Chapman (2017) [29] studied seasonal variations of physical activities and engagement in urban greenspace. Gibbons, Malouf, Spitzberg, Martinez, Appleyard, Thompson, and Tsou (2019) [30] used tweet sentiments to predict residential population health such as mental health, sleep quality, and heart disease at a census tract level, and Zhou, Wang, and Li (2017) [31] collected online reviews on Tripadvisor.com to investigate interactions between tourists and local residents in ten US cities. Photos in Flicker.com have

integrated regional visualizations that map the landscape perception and value for Yosemite Valley, High Line Manhattan, and San Francisco Coit Tower [28]. While these social media studies usually address issues at the regional or city scale, there are not many studies that have focused on public spaces at a site level. A few examples include a landscape character study using Instagram photos and hashtags of Freeway Park in Seattle and a post-occupancy study using Twitter tweets and hashtags for the High Line Park in New York City [22,32]. For platforms such as Tripadvisor and Google Places, the postings are exclusively related to place reviews and ratings, which are important in providing a comprehensive understanding of urban public spaces.

However, few studies have examined the perceived qualities and experiences using location-based social media reviews, which provided essential information on public space usages and attachments. We tackle this gap by conducting a case study using online review data from TripAdvisor to investigate how users comment their experiences in Bryant Park, New York City (NYC), one of the most successful public spaces in the US. Through machine learning and natural language processing techniques, we provide a comprehensive and in-depth analysis of a large amount of textual data, along with a detailed list of the experiential constructs of the park.

## 2. Methodology

### 2.1. Site and Data Collection

Bryant park is a 9.6-acre public park located in Midtown Manhattan. It is known for its seasonal gardens, sculptures, free activities, dining options, restrooms, seating, lawn, and shade, which make it a beloved and comfortable big city park. The high quality of park facilities and vibrant social programs accommodate the needs for public life for residents and visitors during all seasons. The park is owned by the New York City Department of Parks and Recreation and was listed in the National Register of Historic Places in 1966. It is managed by the private not-for-profit organization Bryant Park Corporation who has consistently improved the park over time and has won 74 awards for Bryant Park operations since 1980 [33]. Its long-standing success was widely recognized as a role model for publicly owned, privately managed, and financially self-supporting parks by Gehl Institute and Project for Public Spaces [34,35].

However, as one of the busiest public spaces in the world with 12 million people visiting annually, Bryant Park is rarely studied in academic literature regarding its user experiences and perceived qualities over the past decade. This study takes Tripadvisor review data as a proxy to investigate how visitors use and value Bryant Park as a public space.

With 390 million monthly web visitors and 435 million reviews for 7 million accommodations, restaurants, and other attractions [36], Tripadvisor is the most popular review platform for tourism researchers. However, limitations could exist as Tripadvisor could represent certain socio-personal characteristics in particular age, income, or racial groups. Our dataset has 11,419 Tripadvisor reviews from 10,615 users on the attraction page named "Bryant Park" in New York City. The ratings of these reviews are overwhelmingly positive as 94.8% of reviews have four or five stars and only 0.5% of reviews have one or two stars. As Figure 1 shows, the quantity of reviews rose from 2010 to 2017 and started to fall during the years 2018 and 2019. The winter season (November, December, and January) was the most posted time period of the year. Wednesday was the most posted weekday. A total of 570,284 words were included with an average of 49 words per review. The distribution of review length is shown in Figure 2.

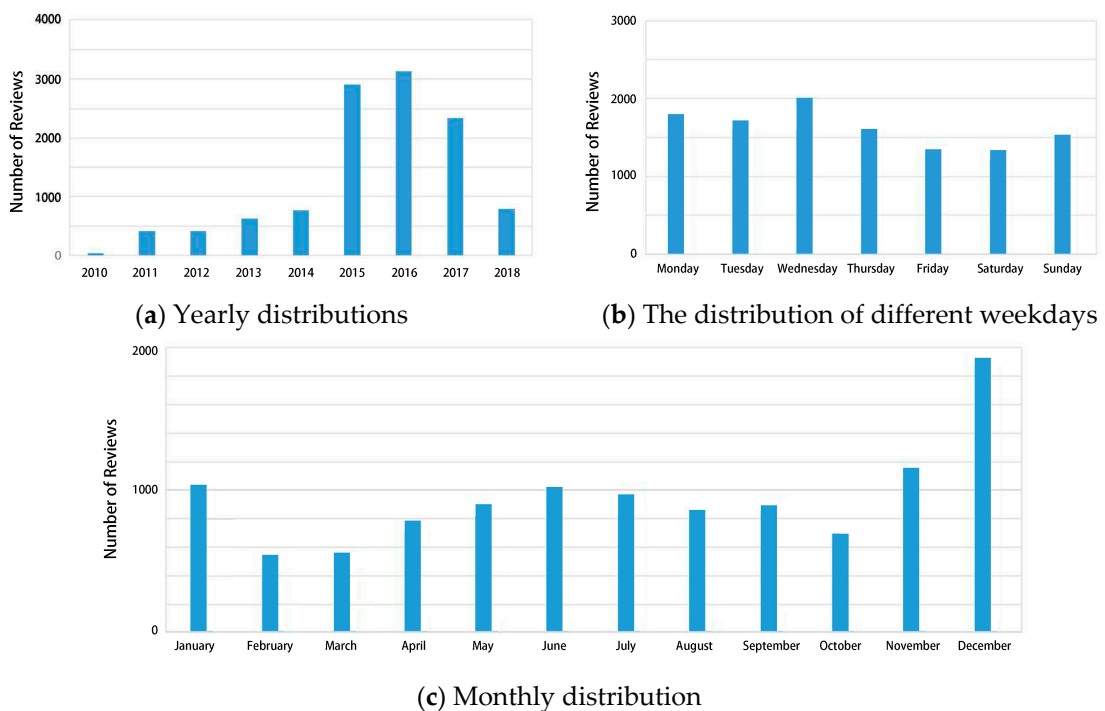

(**a**) Yearly distributions      (**b**) The distribution of different weekdays

(**c**) Monthly distribution

**Figure 1.** Review distributions in different timeframes.

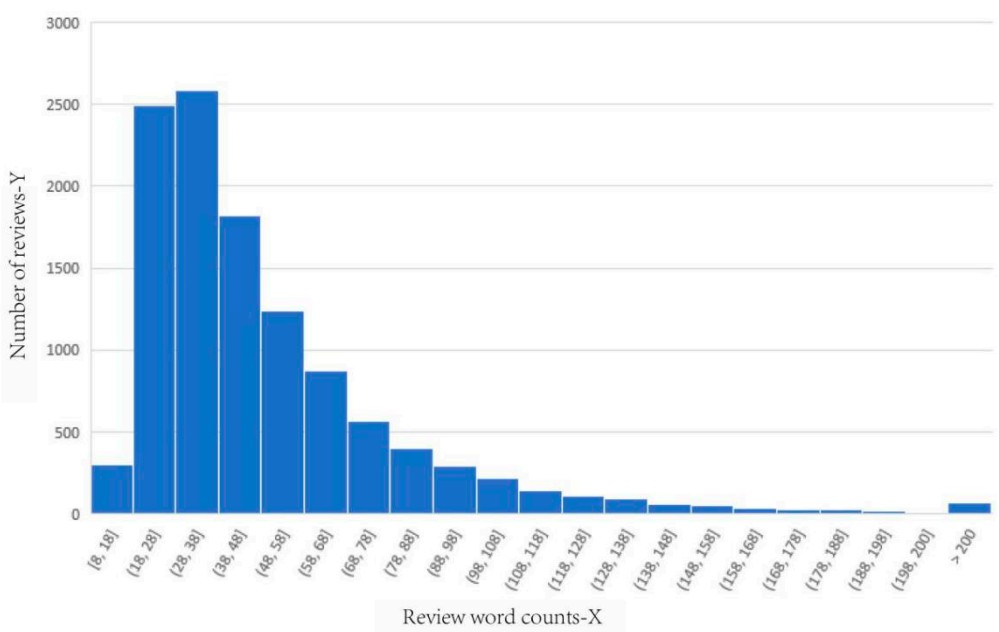

**Figure 2.** Distributions of the review word counts.

## 2.2. LDA (Latent Dirichlet Allocation) Modeling

LDA is one of the most commonly used methods to perform topic modeling tasks. It is a generative machine learning algorithm that can reveal the hidden relationships in a set of review corpus [37]. By utilizing a three-level hierarchical Bayesian model, LDA models each document (review) through a distribution of underlying topics while each topic is represented by a distribution of terms. For large amounts of data, LDA is very efficient to classify documents (reviews) into various topics and calculate the relevance between documents (reviews) and topics. The basic assumption of LDA is that each word of each document (review) is independently extracted from the whole corpus. As Figure 3 illustrates, a topic consists of different topic terms, which are drawn based on the co-occurrence relationships

between terms and documents (reviews). Each term has a term weight to quantify how important the term is in representing the topic. Each review can be described by the composition of multiple topics with their topic weights. For example, Topic #1 dominates Review #2, while Review #1 and Review #3 were composed of a relatively balanced distribution among Topic #1, Topic #2, and Topic #3.

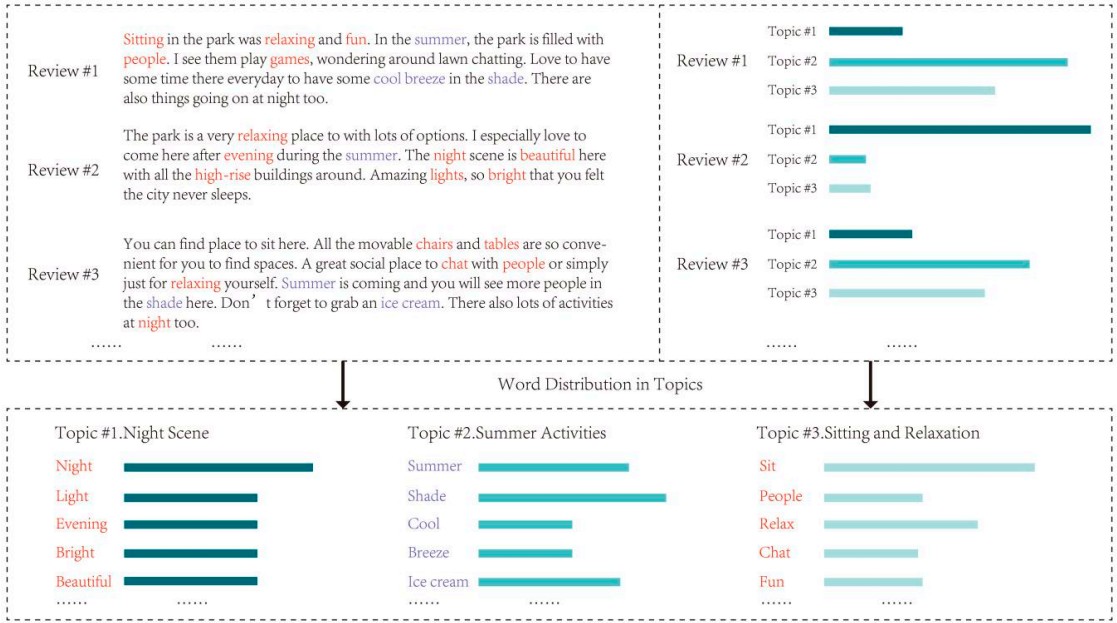

**Figure 3.** Latent Dirichlet Allocation (LDA) diagram.

To evaluate the performance of LDA, this study uses the Cv Coherence measure, which assesses the level of statistical similarity of the topic terms for each topic in the corpus [38]. A Cv Coherence analysis was conducted and the hyperparameter (number of topics) of the LDA model was determined based on the best corresponding Cv Coherence performance.

*2.3. Topic Interpretation*

To have an accurate understanding of the Tripadvisor reviews, we cannot solely rely on the LDA modeling results, which are purely extracted from machine learning calculations. Human interpretations on the semantic context of Topic Terms are needed. Therefore, we conducted a content analysis for Tripadvisor reviews that most related with each LDA topic. Figure 4 showed our general topic interpretation process. After LDA modeling, the top 100 reviews for each topic were first selected based on their topic weights from high to low (a higher topic weight meaning the review is more related with that topic). Then, the authors (three landscape architecture researchers) read all selected 100 reviews and took notes on their understandings of the topic terms that showed up in each review. Finally, we gave each topic a topic name and synthetic description based on our understandings of all the topic terms.

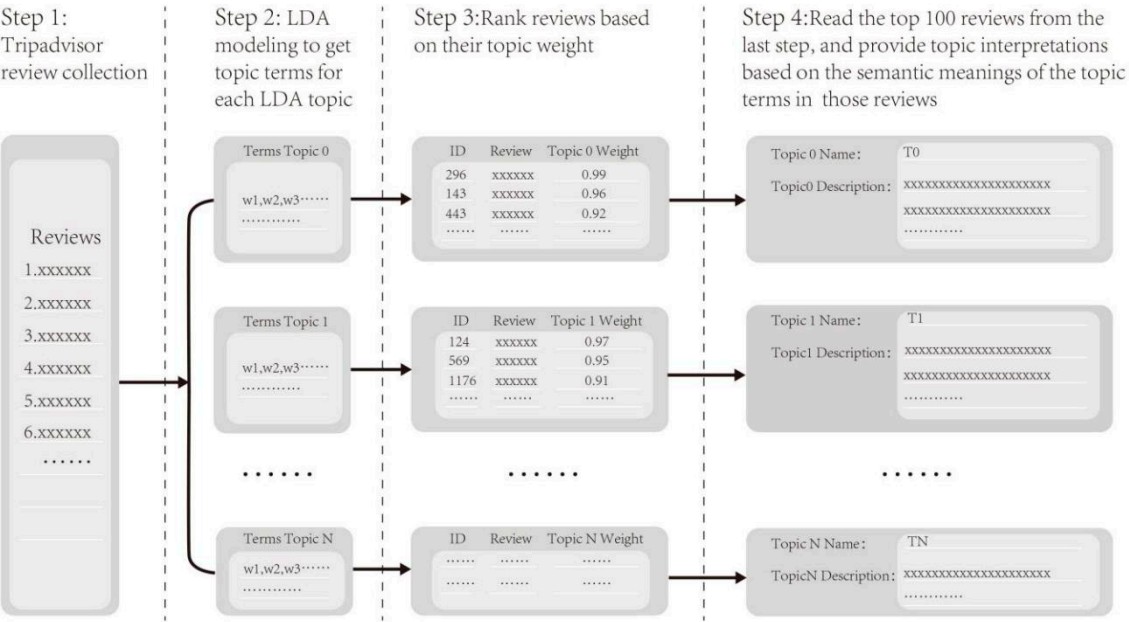

**Figure 4.** The illustration of review interpretation process.

## 3. Results

### 3.1. General Topics and Descriptions

Before generating LDA topics, we conducted a topic Cv Coherence analysis to decide the optimum number of topics. Three rounds of Cv Coherence modeling results were shown in Figure 5 from topic numbers 1 to 29. As the yellow line (average of all three rounds) indicated, a topic number of 5, which had the highest Cv Coherence value, was selected as the hyperparameter for LDA modeling. The resulting LDA topics with their corresponding top 20 topic terms are listed in Table 1. Although most topic terms were different for different topics, words such as "people, day, nice, great" were shared across different topics. Five topics surfaced and were named T0 (Amenities), T1 (Holiday Favorite), T2 (Summer Hotspot), T3 (Place to Relax), and T4 (General Experience). These are major themes that have been mentioned in Tripadvisor reviews, and they generally match our interpretations after reading all top correlated reviews for each topic. The topic names and descriptions for all topics are shown in Table 2.

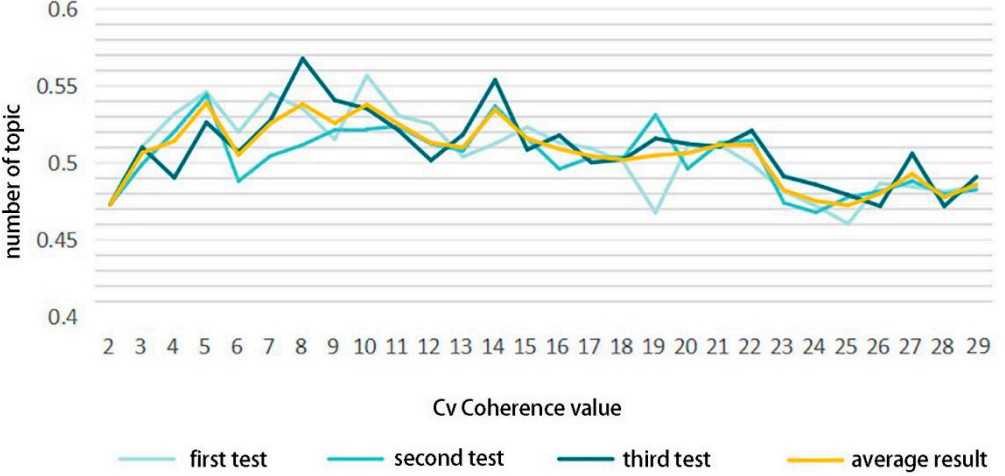

**Figure 5.** Cv Coherence analysis result.

**Table 1.** LDA topics and terms (term weights * topic terms).

| Amenities (Topic 0)- | Holiday Favorite (Topic 1) | Summer Hotspot (Topic 2)- | Place to Relax (Topic 3)- | General Experience (Topic 4)- |
|---|---|---|---|---|
| 0.046 * "library" | 0.042 * "ice" | 0.021 * "summer" | 0.030 * "sit" | 0.022 * "day" |
| 0.032 * "public" | 0.036 * "skating" | 0.021 * "free" | 0.030 * "great" | 0.015 * "time" |
| 0.017 * "table" | 0.030 * "rink" | 0.021 * "movie" | 0.025 * "nice" | 0.014 * "would" |
| 0.013 * "area" | 0.030 * "Christmas" | 0.013 * "people" | 0.025 * "people" | 0.011 * "lovely" |
| 0.013 * "behind" | 0.022 * "shop" | 0.012 * "time" | 0.020 * "watch" | 0.010 * "went" |
| 0.012 * "beautiful" | 0.018 * "great" | 0.011 * "event" | 0.018 * "relax" | 0.010 * "really" |
| 0.011 * "carousel" | 0.016 * "winter" | 0.010 * "night" | 0.016 * "coffee" | 0.009 * "visit" |
| 0.011 * "chair" | 0.015 * "market" | 0.010 * "great" | 0.015 * "time" | 0.009 * "people" |
| 0.010 * "well" | 0.015 * "food" | 0.010 * "game" | 0.014 * "enjoy" | 0.009 * "night" |
| 0.010 * "right" | 0.014 * "time" | 0.009 * "show" | 0.013 * "lunch" | 0.009 * "visited" |
| 0.009 * "also" | 0.013 * "holiday" | 0.008 * "always" | 0.013 * "take" | 0.009 * "back" |
| 0.008 * "small" | 0.013 * "skate" | 0.008 * "activity" | 0.011 * "around" | 0.008 * "building" |
| 0.008 * "next" | 0.011 * "little" | 0.008 * "playing" | 0.011 * "food" | 0.008 * "loved" |
| 0.008 * "free" | 0.011 * "tree" | 0.008 * "nyc" | 0.011 * "little" | 0.008 * "around" |
| 0.008 * "book" | 0.010 * "stall" | 0.008 * "going" | 0.011 * "drink" | 0.008 * "enjoyed" |
| 0.007 * "reading" | 0.009 * "visit" | 0.007 * "chess" | 0.010 * "table" | 0.007 * "beautiful" |
| 0.007 * "game" | 0.009 * "fun" | 0.007 * "play" | 0.010 * "eat" | 0.007 * "walked" |
| 0.007 * "clean" | 0.009 * "nice" | 0.007 * "yoga" | 0.009 * "lovely" | 0.007 * "came" |
| 0.007 * "green" | 0.008 * "around" | 0.007 * "music" | 0.009 * "day" | 0.007 * "nice" |
| 0.007 * "lawn" | 0.007 * "vendor" | 0.007 * "concert" | 0.009 * "middle" | 0.007 * "little" |

*3.2. Topic Distributions*

For the temporal variations of LDA topics, we aggregated all weights for each topic and took the average of different weekdays and months to show the distributions of topic mentions through time. As Figure 6 shows, each topic has its own best weekday as no weekday was the most mentioned for all topics. From Figure 7, T4 (General Experience) and T2 (Summer Hotspot) were both most mentioned on Friday, although the differences between Tuesday, Friday, and Saturday for T4 (General Experience) were minor. No topics had Monday as their most mentioned weekday. Regarding monthly distributions, T0 (Amenities) and T3 (Place to Relax) were highly mentioned in a wide range of months from April through September. T2 (Summer Hotspot) presented gradual growth beginning in January and peaking in August with a concentration in summer months (June–September). T1 (Holiday Favorite) was most active during the winter season and least between April through October. T4 (General Experience) had the least variation throughout the year as its average topic weight did not fluctuate as largely as other topics.

**Table 2.** Topic descriptions and examples.

| Topic Name | Description | Review Examples |
|---|---|---|
| Amenities (Topic 0) | This theme focuses on passive activity, relaxation, and the site features that enable and encourage people to take a seat and stay awhile. The theme characterizes the park as a place to take in the world- the sun, the people, the food, and the beautiful surroundings. People-watching and leisure are passive activities that occur here, and food often provides comfort and rejuvenation while doing so. | The park contains the New York **public library**, a restaurant, several street shops, board **game**s, and a huge **lawn area**. |
|  |  | Plenty of seating, activities, eating options and a **carousel** that plays french music....ahh, very relaxing. |
|  |  | Take a visit to the **public library** then rest up among the trees of the park or stretch out in the **lawn**. |
|  |  | This park is **beautiful**, well kept, options for **game**s and some recreational activities like **table** tennis, even **free** open WiFi. |
|  |  | Lovely **small** park **next** to the **library** in Manhattan. |
| Holiday Favorite (Topic 1) | The winter holiday season is featured in this theme with a focus on seasonal shops, specialty food vendors, and ice skating. The spirit of the holidays is enhanced by the seasonal décor, the tree, and the many programmed features such as the Winter Village. The site is a favorite among many holiday goers for its fun and lively atmosphere, despite the crowds. Food vendors sell a unique variety of treats, and shopping features novelty and crafted items appropriate for special gifts. | From the ice **staking rink** to the pub, to the **Christmas** decorations and huge **tree**-the whole atmosphere was terrific. |
|  |  | All the **market stalls** and artisans were so much **fun** to **visit** and talk to. I'm someone who is full of questions and loves to examine well-crafted items; the owners were all so patient and kind, and had loads of suggestions for me. |
|  |  | With over 140 **shop**ping kiosks, **food vendor**s, a carousel, **ice skating rink** and beautiful **Christmas tree** there is something for everyone |
| Summer Hotspot (Topic 2) | This topic centers on enjoyable summer activities and the vibrance of the park during the warm weather months. Site programming and events including movies, Broadway shows, outdoor movies, and yoga are highly recommended, as are the free games such as chess and ping pong. The park provides affordable and amusing activities throughout the season which encourage users to relax, people-watch, and to eat and drink. | It's a gem! It has everything! **Summer yoga** classes, outdoor **movie night**s, **chess**, bingo, dance and **music** sessions! |
|  |  | Love, love this park! Been here before, and there's food and every Friday **night** this **summer** there's **free play** or entertainment. There was a Shakespeare **play** happening at this **time**. Chairs all set up and **free**! Very cool! Just a cool place to visit. |
|  |  | My family and I stopped at Bryant Park late Friday afternoon and to our delight there was a lot **going** on! Between the jazz **concert**, food, beer, **game**s, miniature golf, **chess** and lots more, a stroll in the park turned out to be fun in the park. My favorite was the Jazz **concert** and the **game**s. |
|  |  | Monday **night**s during the **summer**, pack a blanket, bottle of wine, repellent, and head out to Bryant park for classic **movie**s. **Movie** begins at sunset with a Warner Bros cartoon as appetizer, which by itself can melt your heart. |

**Table 2.** *Cont.*

| Topic Name | Description | Review Examples |
|---|---|---|
| Place to Relax (Topic 3) | This theme focuses on passive activity, relaxation, and the site features that enable and encourage people to take a seat and stay awhile. The theme characterizes the park as a place to take in the world- the sun, the people, the food, and the beautiful surroundings. People-watching and leisure are passive activities that occur here, and food often provides comfort and rejuvenation while doing so. | **Great** park in the city to hang out and **relax** after shopping or site seeing. Plenty of seats spread all around, **great** friendly atmosphere with **food** easily to obtain from surround outlets to **eat** in the park. Activities always going on to **watch**. A **great** few hours break. |
| | | In the heart of midtown Manhattan in the **great**est city in the world, come feel the beat! An open air park, depending on the **day**, vendors, **food** and entertainment. I like the noon rush with workers pouring onto the streets and through the park. Bring the family for a short visit or **people watch** for hours. |
| | | A big beautiful park in the **middle** of Manhatten. Lots of places around to get **food/drink**s. Then lay down on the grass, **enjoy** the sun or all the **people** and just let time pass you by and recharge. |
| | | Right in the centre of midtown Manhattan is this beautiful garden, flanked by restaurants and bars and **coffee** shops. we sat in the sun and had a cold beer whilst **watch**ing the world go by. A **nice** spot to **take** a break in the city. |
| | | It's a park ok, trees, grass, plants, places to **eat** and **table**s around. But. I love it. It's not a place to hurry and **take** photos, it's better to go with **time**, buy something to **eat** and **drink** (picnic perhaps) and stay for a while. Peaceful place in a crazy city. |
| General Experience (Topic 4) | This topic provides a general concept as a casual and typically pleasant experience. Many users in this category stumbled upon the park or strolled through it based on its convenient location. While some characterized the park as an ordinary typical urban park, it still serves as a respite from the hustle and bustle of the city and a place to visit again and again. The attractiveness of the park, good atmosphere, and buzzing ambiance both day and night add to this park's appeal. Additionally, whether positive or negative, in this topic users compare the location to Central Park. | Bryant Park is quite a **nice** place to sit, enjoy the view and the sun. A very **beautiful** park to get to know in NY. There's this restaurant (seems **nice**) inside the park, which seems a great option to spend some more **time** in this **beautiful** place. |
| | | This is a **really lovely** area. Lots of places to sit and watch the world go by. We had a **nice** meal in one of the restaurants. Lots and lots of seating and snack bars or you could buy food elsewhere and just grab a table. A **nice** place to chill. |
| | | It was a **lovely day** to hang out in the park and get a breath of fresh air. Bryant Park offers a **nice** setting with plenty seating to grab a bite, coffee or drinks and hang out with friends or just to relax or **people** watch. |
| | | I had a relaxing **time** walking **around** and sitting at Bryant Park. It's centrally located so we did so after some shopping at Macys which was walking distance from there. There were some **nice** coffee shops across the road from the Park. Overall it was very pleasant and it was a sunny **day** too. |
| | | Lots of tables, lots of benches, lots of birds and lots of **people**. This is literally in the center of the city. A **nice** place to sit and relax after walking 5th Ave or **visit**ing Times Square. |

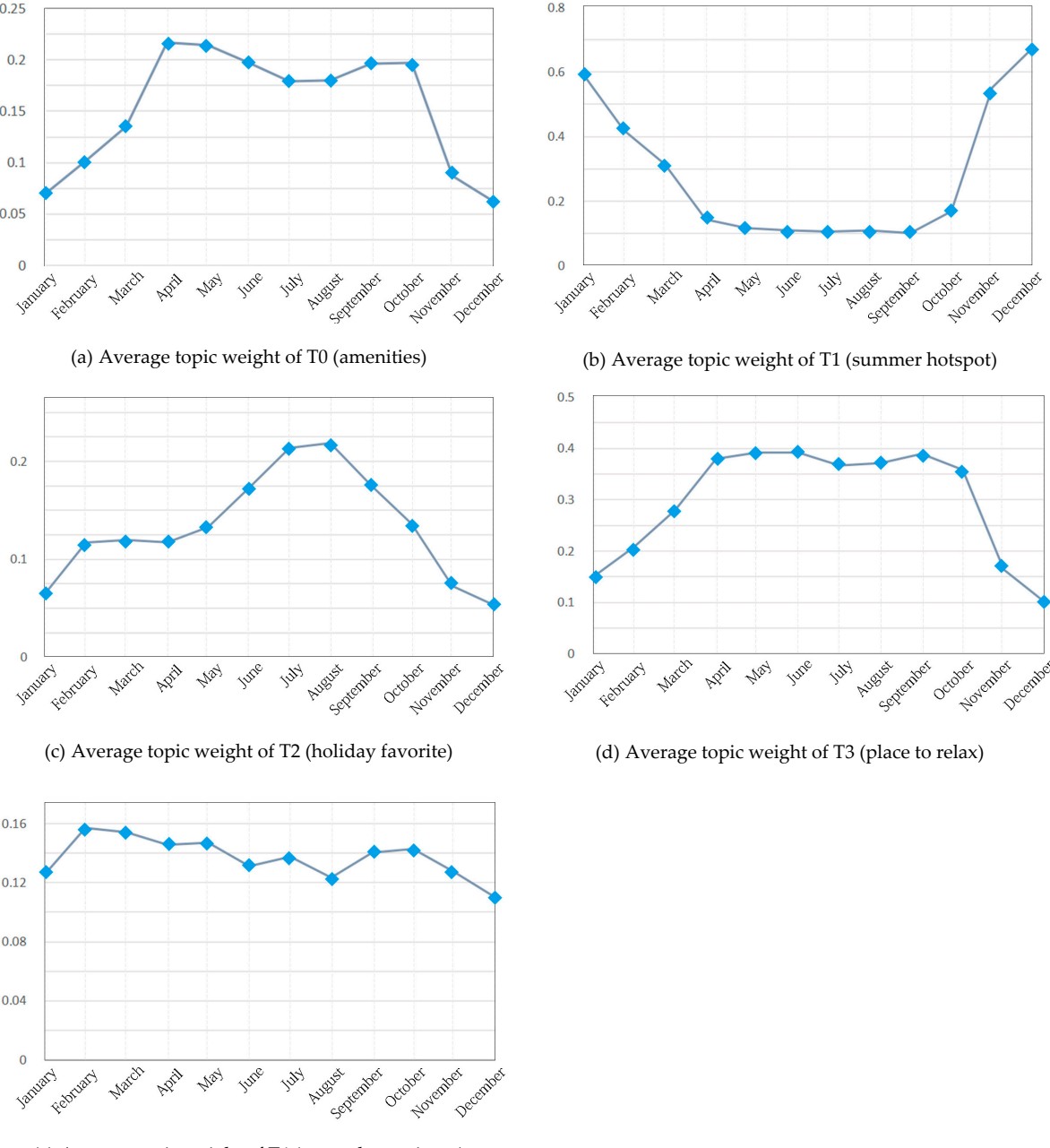

(a) Average topic weight of T0 (amenities)

(b) Average topic weight of T1 (summer hotspot)

(c) Average topic weight of T2 (holiday favorite)

(d) Average topic weight of T3 (place to relax)

(e) Average topic weight of T4 (general experience)

**Figure 6.** Average topic weights for different weekdays.

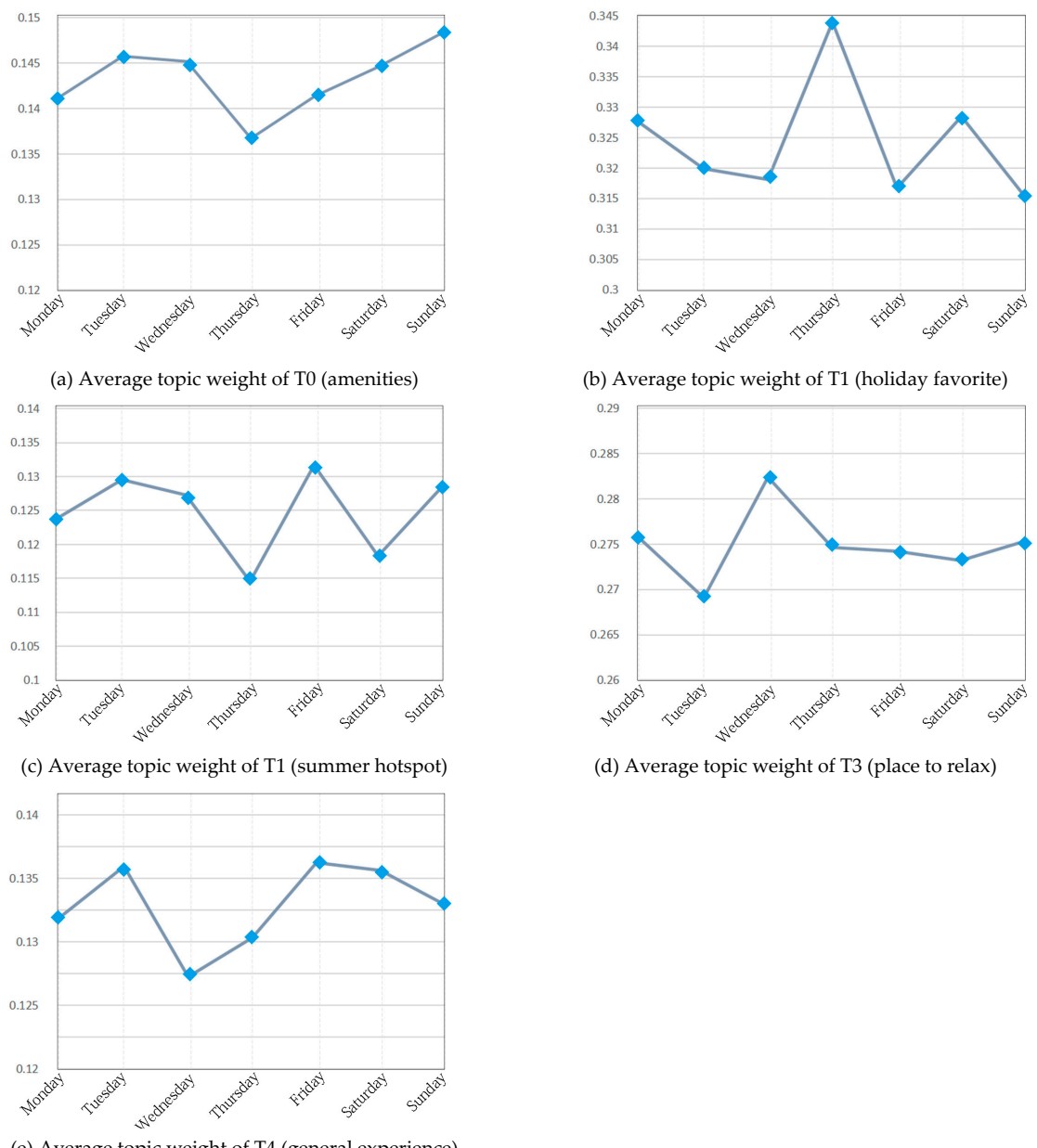

(a) Average topic weight of T0 (amenities)

(b) Average topic weight of T1 (holiday favorite)

(c) Average topic weight of T1 (summer hotspot)

(d) Average topic weight of T3 (place to relax)

(e) Average topic weight of T4 (general experience)

**Figure 7.** Average topic weights for different months.

## 4. Conclusions and Discussion

### 4.1. Public Space Experiences in Bryant Park

As more people live in cities, knowledge on how to make cities livable becomes a significant challenge. The need for public spaces is increasing, given the scarcity of land [39]. This study presents a social media method to investigate the perceived site quality and experiences in a NYC public space, Bryant Park. Five topics were detected from 11,419 Tripadvisor reviews, including T0 (Amenities), T1 (Holiday Favorite), T2 (Summer Hotspot), T3 (Place to Relax), and T4 (General Experience). These topics reflect an encompassing picture of what Bryant Park is and reveal an overall idea of park perceptions in sheer user perspectives.

In general, users wrote about what they saw, what they did, and how they felt during the visit. T0 (Amenities) listed some key spaces, facilities, and things related to park experiences in topic terms such as table, carousel, chair, book, game, lawn, restaurant/food, building, tree, bathroom/restroom,

and flower. Contextual elements such as the public library and streets are also a significant part of this topic, indicating that having a desirable location with great accessibility could also be important for park users.

T3 (Place to Relax) depicted a classical Bryant Park scene with topic terms like sit, people, watch, relax, coffee, lunch, food, drink, table, and walk. Food, seating, and trees were found to be the key elements for successful social places to attract people [19]. During the time of this study, Bryant park did a fantastic job of incorporating all three into the park, most likely positively impacting the site user experience. The food kiosks, the lodge, and holiday shops provide various choices for eating and drinking. The park's 1000 movable chairs have become one of the icons of this destination [34]. The well-maintained lawn and planting beds under the monoculture of hundreds of London Plane Trees create a significant greenery setting which facilitates restoration from attentional fatigue by the fast-paced lifestyle and density of NYC [25]. In addition, over 10,000 summer annual plantings and 17,000 spring bulbs planted each year make the park full of surprises year round. T4 (General Experience) consists of many general terms for park experience. Terms like day time, lovely or loved, enjoyed, beautiful, nice, night, and evening are mentioned in most reviews across our dataset. We also see overwhelmingly positive language on the reviews in this topic, indicating how Bryant Park could contribute to the well-being of the public.

The programming and events planning of public space is as important as the physical design [40]. Since Bryant Park is run by Bryant Park Corporation (BPC), which is a private management company, its operations could leverage private sector techniques and management methods to drive competition and innovation on site programming. Our T1 (Holiday Favorite) and T2 (Summer Hotspot) topics show how visitors engage with the various programmed social activities of the park. In T1 (Holiday Favorite), winter holidays are featured with topic terms including ice skating rink, Christmas tree, shops, food, and market. Many holiday decorations, temporary installations, and programmed features make the park a favorite of holiday goers. During the summer, different activities were highlighted in T2 (Summer Hotspot) with topic terms such as movie, event, game, show, chess, yoga, music, concert, ping pong, and film. Some interactions between different topics are apparent since they rarely occur independently. For example, food-related terms such as restaurant, food, drink, coffee, and evening were shown across all key topics T0, T1, T2, and T3. This indicates food is an essential element of Bryant Park experiences.

An advantage of utilizing online reviews is the long timeframe of review information. Since our data ranged from 2010 to 2019, we can analyze the temporal dimensions of each LDA topic. Firstly, the positive experiences in Bryant Park showed sustained patterns as T4 (General Experience) presents fewer fluctuations in both weekly (Figure 7e) and monthly (Figure 6e) charts. This continuity is quite an achievement since other public spaces such as Freeway Park in Seattle have shown strong seasonal variations in park attachment perceptions [32]. Secondly, weekdays are more important than weekends as an everyday respite function for Bryant Park. NYC Manhattan has a large number of commuting workers every day. Work stress and anxiety are a natural mental burden for many people. Studies have found that Tuesday is both the most productive and stressful day of the week for workers [41,42]. As a function of mental restoration, T3 (Place to Relax) was mentioned most in our results on Wednesdays during the middle of the work week. Thirdly, more usage of general park amenities may be expected on weekends and during warmer seasons. This is reflected in topic T0 (Amenities), which are most mentioned on Sundays and months between April through October.

### 4.2. Online Reviews for Public Space Research

Although social media data is gaining significant attention in social science research, it is not without limitations. One large obstacle is that such data are inherently messy as they are not gathered using a systematic and statistically guided methodology. Fake accounts, missing data, and poor classifications can be common issues [43]. Nevertheless, online review data on Tripadvisor are different than other social media sites. It has a highly organized and structured filtering system where users only

post information about their trip experiences. Fake reviews for hotels or restaurants may exist because of business interests. In this study, we did not find any irrelevant or inappropriate information when reading reviews for our content analysis in this study. With some caution, we think online review data in Tripadivisor for public spaces like Bryant Park have high quality and could assist the understanding of user experiences.

Before the use of the automobile, there was significant activity on most streets or open spaces of the city. Currently public spaces, regardless of their quality, are one of the primary facilitators of city life and social interaction. Because of the freedom the automobile provides, our built environments and society as a whole are physically scattered and isolated since we can drive to further places and connect through the internet [21]. Time spent in public spaces is often leisure time and highly dependent on the qualities and experiences of the place [35]. Therefore, public space managers or developers need to be more well-prepared and nimbler to meet the changing and diverse needs of city life. This study shows the potential for using online reviews to understand and track the perceptions of park users. Our topics and topic terms are the results of a large sample group of 10,615 users, which are helpful to weigh contrary interests and to supplement single expert views [44]. They could give a more comprehensive and equitable representation of place perceptions. They could also bring researchers closer to the public and assist the understanding of cultural ecosystem services in urban areas. For urban designers and landscape architects, our study may promote human-centered and evidence-based design in lieu of following design trends of aesthetic fad [45]. Local businesses, community activists, and real estate developers could use our findings to better evaluate their properties and think about how to leverage the nearby park to activate their communities. For example, our T1 (Holiday Favorite), which includes topic terms such as shop, market, stall, vendor, and food, indicates that temporary vendors or food services could be great placemaking programs for local retailers to organize their businesses in or around the park. Planners and city administrations could also gain insights about public spaces and propose future projects for communities that lack public amenities like Bryant Park.

### 4.3. Limitations

This study is subject to several limitations. First, potential selection bias about demographics still exists [46]. Bryant Park visitors include both local residents and travelers who may view the park differently. Non-English reviews were not considered in our studies though people from around the world are visiting the park. Future studies could use surveys or interviews that target the demographics our study may have missed to mitigate the biases that reside in our results. Second, our results only target the LDA topics and their topic terms. The topic terms could be further coded and categorized based on their semantic context in those highly associated reviews. Therefore, more specific concepts with design implications that are embedded in our LDA topics could be identified. Third, the LDA methodology only captures highly mentioned terms in our corpus. Some emerging or niche concepts in our datasets could not be detected. It is also possible that not all topics have research and practice implications (in this case topic 4, General Experience). Future studies could conduct multiple LDA analysis based on different seasons, review ratings, or specific weekdays to gain a finer understanding of user perceptions of place. It is important to further examine our topics by looking at the complex interactions of terms between different topics and how those concepts connect with each other.

**Author Contributions:** Conceptualization, T.W. and Y.S.; methodology, Y.S.; software, J.F. and Y.S.; validation, J.F. and Y.S.; formal analysis, Y.S.; investigation, T.W.; writing—original draft preparation, Y.S. and T.W.; writing—review and editing, Y.S. and T.W.; visualization, T.W.; supervision, Y.S.; project administration, T.W.; funding acquisition, T.W. All authors have read and agreed to the published version of the manuscript.

**Funding:** This research received no external funding.

**Acknowledgments:** We would like to thank Cheng Zhang and Dahao Li for their support and the reviewers for their professional comments and suggestions.

**Conflicts of Interest:** The authors declare no conflict of interest.

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
