# Peer review of "Understanding Perceived Site Qualities and Experiences of Urban Public Spaces: A Case Study of Social Media Reviews in Bryant Park, New York City"

_sustainability, doi:10.3390/su12198036_

Round 1
Reviewer 1 Report
This is an interesting and important research that investigates the ways people perceive and experience urban public space using emerging big/open data online. The paper is largely articulate, with a clear and coherent structure, however, it embodies key issues of research design and writing. My decision for the current manuscript is major revision. I hope the authors could take the following comments/suggestions positively and I am looking forward to reading the revised article.
1. While this research applied LDA modelling to analyse a large number of TripAdvisor reviews, human interpretation of text still plays a vital role. Three landscape architecture researchers (not sure from the inside or outside) were only recruited to assist the interpretation of topics/words when LDA modelling was completed. It would be much better if this could happen from the beginning – the content analysis that leads to the basic rules for LDA modelling.
2. If the goal of the paper is to show how well the analysis of online reviews on urban public space could inform the landscape/urban design practice, then the classification of topics should be more geared towards practice with a clearer structure. The existing paper identified five key topics comprising ‘amenities’, ‘holiday favourite’, ‘summer hotspot’, ‘place to relax’, and ‘general overview’. Some are more related to activities while some are more related to time, and the last category ‘general overview’ hardly contributes to any useful implication for research and practice. I would strongly suggest that the topic ‘general overview’ should be replaced by several more meaningful terms. Further, it would be much better if the key topics could revolve around those aspects that are critical in the design practice (e.g. terms related to the physical, socio-cultural, and economic attributes of the park) while being difficult to measure or analyse through conventional methods such as field studies.
3. Whatever the classification of topics would be, there are complex interrelations between different topics; they rarely exist independently. For instance, proximity to restaurants could be related to two or more topics such as ‘amenities’ and ‘holiday favourite’. While the analysis of such complex interrelations should be best addressed in a new paper, it needs to be at least briefly mentioned in this manuscript.
4. The research gap should be summarised by one or two sentences in the abstract and introduction: what are missing in the existing literature and practice? How these gaps would be tackled in this research?
5. Limitations of the research using emerging big/open data online should be specified earlier in the section of methods, and another key component here should be introducing how this research could mitigate the negative influences of such limitations. Further, it is necessary for this research to mention the potential bias 1) due to people’s identity as local residents or travellers and 2) towards the English language. For the case of Bryant Park, local residents and travellers may have quite different experiences. Also, non-English reviews were not considered while Bryant Park is visited by people with diverse socio-cultural backgrounds.
6.The writing of the paper could be improved on the following three aspects. First, be more specific about key proofs or viewpoints. For example, in [line 92-94] “A few examples include a landscape character study using Instagram photos of Freeway Park in Seattle and a post-occupancy study using Twitter for the High Line Park in New York City”, it is necessary to point out what content of Twitter is used for the related research: text, image, or both? Second, some sentences seem to be wordy and convoluted. For example, [line 31-32] “World Health Organization (WHO) released guidelines to use built environment approaches that use park and recreation and natural areas as interventions to promote physical activity” could be simplified to “World Health Organization (WHO) suggested that open or green space could assist to encourage physical activities”. Third, typos should be removed. For example, in line 101 “one of the most successful public spaces in the US according to?”, the question mark is a clear typo and a reference is seemingly missing here. Please review the manuscript carefully to clear all such issues; bear in mind that I could not identify all of them for you.
Good luck with editing!
Author Response
This is an interesting and important research that investigates the ways people perceive and experience urban public space using emerging big/open data online. The paper is largely articulate, with a clear and coherent structure, however, it embodies key issues of research design and writing. My decision for the current manuscript is major revision. I hope the authors could take the following comments/suggestions positively and I am looking forward to reading the revised article.
Thank you. We appreciate your comments and we are glad to provide a revised article, and we’ve highlighted our changes in the newly submitted manuscripts.
- While this research applied LDA modelling to analyse a large number of TripAdvisorreviews, human interpretation of text still plays a vital role. Three landscape architecture researchers (not sure from the inside or outside) were only recruited to assist the interpretation of topics/words when LDA modelling was completed. It would be much better if this could happen from the beginning – the content analysis that leads to the basic rules for LDA modelling.
Thanks for the comments. We agree that human interpretation plays a vital role in our work. The process of grouping topic terms is conducted through LDA algorithm and fully automatic. Human interpretations didn’t take place until we completed LDA running. However, all three researchers (inside) did followed the whole research process of this paper, and they are trained and aware of the steps and the rationales behind our analysis. To point out this part, we added new content at Line 161 “researchers(inside) who were trained and have followed the whole research process”.
- If the goal of the paper is to show how well the analysis of online reviews on urban public space could inform the landscape/urban design practice, then the classification of topics should be more geared towards practice with a clearer structure. The existing paper identified five key topics comprising ‘amenities’, ‘holiday favourite’, ‘summer hotspot’, ‘place to relax’, and ‘general overview’. Some are more related to activities while some are more related to time, and the last category ‘general overview’ hardly contributes to any useful implication for research and practice. I would strongly suggest that the topic ‘general overview’ should be replaced by several more meaningful terms. Further, it would be much better if the key topics could revolve around those aspects that are critical in the design practice (e.g. terms related to the physical, socio-cultural, and economic attributes of the park) while being difficult to measure or analyse through conventional methods such as field studies.
Thanks for the comments. We agree that the Topic 4 “general overview” provided few implications for research and practice. However, we think it is still necessary to present the result of Topic 4 since it is one of themes from reviews. Because our data are directly collected from users who write reviews in an open manner, not all themes they say will have research and practice implications. We think this is a limitation of LDA method and social media methodology. We’ve added new content at Line 295 “It is also possible that not all topics have research and practice implications (in this case topic 4). ” We also changed all the name of topic 4 as “general experiences” in the manuscripts.
Regarding to have key topics revolve around the aspects that are critical to design practice, we think this could be a great opportunity for future studies to dive into each topic term and find out their context and meanings for design implications. We’ve added new content at Line 293 “with design implications”.
- Whatever the classification of topics would be, there are complex interrelations between different topics; they rarely exist independently. For instance, proximity to restaurants could be related to two or more topics such as ‘amenities’ and ‘holiday favourite’. While the analysis of such complex interrelations should be best addressed in a new paper, it needs to be at least briefly mentioned in this manuscript.
We agree with the idea to address the complex interactions and specifics could be a new paper and we plan for it as a future study. New content was added at Line 238 “Some interactions between different topics are apparent since they rarely occur independently. For example, food-related terms such as restaurant, food, drink, coffee, and evening were shown across all key topics T0, T1, T2, T3. This indicates food is an essential element of Bryant Park experiences.” and at Line 298, “It is important to further examine our topics by looking at the complex interactions of terms between different topics and how those concepts connect with each other.”
- The research gap should be summarised by one or two sentences in the abstract and introduction: what are missing in the existing literature and practice? How these gaps would be tackled in this research?
Thank you for this comment. We agree with the reviewer. We changed the sentence at Line 96 to explicitly mention what is missing in existing literature and how to tackle the gap “However missing in the existing literature are studies that systematically examine the use of user generated social media reviews which we believe include important content to inform site qualities and experiences. We tackle this gap by conducting a case study using online review data from TripAdvisor to investigate the perceived site quality and experiences in Bryant Park, New York City (NYC) which is one of the most successful public spaces in the US”. We also added and modified the content in abstract to following “Existing literature lacks comprehensive examinations of site qualities and experiences. This study tackle this gap by introducing a case study that uses social media reviews (Tripadivisor.com) to understand the perceived site quality and experiences of Bryant Park in New York City.”
- Limitations of the research using emerging big/open data online should be specified earlier in the section of methods, and another key component here should be introducing how this research could mitigate the negative influences of such limitations. Further, it is necessary for this research to mention the potential bias 1) due to people’s identity as local residents or travellers and 2) towards the English language. For the case of Bryant Park, local residents and travellers may have quite different experiences. Also, non-English reviews were not considered while Bryant Park is visited by people with diverse socio-cultural backgrounds.
Thank you, we agree with these comments. We specifically stated the limitations of using emerging big/open data in the methodology section at lines 121, “However, limitations could exist as Tripadvisor could represent certain socio-personal characteristics in particular age, income or racial groups.” However, we could not mitigate this limitation in this paper so we mentioned in the limitation section to propose future studies on this at lines 289, “Future studies could use surveys or interviews that target the demographics our studies may have missed to mitigate the biases that reside in our results here ” For the potential bias, we agree with the reviewer. We explicitly stated them at line 286 “potential selection bias about demographics still exists [46]. Bryant Park visitors include both local residents and travelers who may view the park differently. Non-English reviews were not considered in our studies though people from around the world are visiting the park”
6.The writing of the paper could be improved on the following three aspects. First, be more specific about key proofs or viewpoints. For example, in [line 92-94] “A few examples include a landscape character study using Instagram photos of Freeway Park in Seattle and a post-occupancy study using Twitter for the High Line Park in New York City”, it is necessary to point out what content of Twitter is used for the related research: text, image, or both? Second, some sentences seem to be wordy and convoluted. For example, [line 31-32] “World Health Organization (WHO) released guidelines to use built environment approaches that use park and recreation and natural areas as interventions to promote physical activity” could be simplified to “World Health Organization (WHO) suggested that open or green space could assist to encourage physical activities”. Third, typos should be removed. For example, in line 101 “one of the most successful public spaces in the US according to?”, the question mark is a clear typo and a reference is seemingly missing here. Please review the manuscript carefully to clear all such issues; bear in mind that I could not identify all of them for you.
Thanks for the comments. We agree with the reviewer. For the first aspect, we added more specifics at Lines 91 to explicitly state the data the studies are using “A few examples include a landscape character study using Instagram photos and hashtags of Freeway Park in Seattle and a post-occupancy study using Twitter tweets and hashtags for the High Line Park in New York City”. For the second aspect, we agree the sentence is wordy and we made the adjustments accordingly at Line 31. For the third aspect, we agree those are type issues from when we transferred a Google doc to word. We’ve made adjustments on those errors at multiple places of the newly submitted manuscripts .
Good luck with editing!
We would thank the reviewer again for your time to provide valuable feedback to make our paper better.
Best Wishes.

Reviewer 2 Report
The article is a study of semantic sentiment analysis from the scrapping of opinions about a New York park (Bryan Park) on TripAdvisor. The methodology is clearly described and the study, without a doubt, has scientific rigor, explained in the procedures and methods of data collection.
However, two improvements are proposed: First, the LDA topics outlined in table 1 can be better included in a Dataset. MDPI has a dataset file system, which improves the interoperability of the manuscript and the clarity of the manuscript. Also, figure 7 is not explained.
On the other hand, perhaps the study sins of descriptive excess and does not enter into a discussion of the meaning of semantic valuations. In what way do the opinions influence the perceptions of the spaces? In what way can the companies in the environment of the site be influenced economically and socially by the opinions in TripAdvisor? How can these opinions put in value not only the revaluation of the surroundings, but all the social structures of the area? I think it is important to discuss these aspects, as they are the ones that will provide other researchers with the keys to the practical implications of this study.
Author Response
The article is a study of semantic sentiment analysis from the scrapping of opinions about a New York park (Bryan Park) on TripAdvisor. The methodology is clearly described and the study, without a doubt, has scientific rigor, explained in the procedures and methods of data collection.
Thank you. We appreciate your comments and we are glad to provide a revised article. We’ve highlighted our changes in the newly submitted manuscripts.
However, two improvements are proposed: First, the LDA topics outlined in table 1 can be better included in a Dataset. MDPI has a dataset file system, which improves the interoperability of the manuscript and the clarity of the manuscript. Also, figure 7 is not explained.
Thanks for the comments. We agree with the reviewer. We are planning to include Table 1 in the dataset through the MDPI dataset file system. We also indicated the descriptions of Figure 7 at Line 189.
On the other hand, perhaps the study sins of descriptive excess and does not enter into a discussion of the meaning of semantic valuations. In what way do the opinions influence the perceptions of the spaces? In what way can the companies in the environment of the site be influenced economically and socially by the opinions in TripAdvisor? How can these opinions put in value not only the revaluation of the surroundings, but all the social structures of the area? I think it is important to discuss these aspects, as they are the ones that will provide other researchers with the keys to the practical implications of this study.
Thanks for the comments. We agree with the reviewer. We added more content at Lines 279 to further the discussions on the value of our findings “For urban designers and landscape architects, our study may promote human-centered and evidence-based design in lieu of following design trends of aesthetic fad [45]. Local businesses, community activists, and real estate developers could use our findings to better evaluate their properties and think about how to leverage the nearby park to activate their communities. Planners and city administrations could also gain insights about public spaces and propose future projects for communities that lack public amenities like Bryant Park.“
We would thank the reviewer again for your time to provide valuable feedback to make our paper better.
Best Wishes.
Round 2
Reviewer 1 Report
I am largely satisfied with the revised article and think that the manuscript would be ready for publication once the following comments are followed and/or considered.
To be followed:
- The research gap is still murky, in particular the related sentence in the abstract (line 14): “Existing literature lacks comprehensive examinations of site qualities and experiences”; this is simply too broad. How about: “there is a lack of studies that investigate how people use and perceive urban parks using quantitative analysis of location-based social media reviews”? Accordingly, line 96 to 100 should be revised to clarify the research gap.
- In figure 3, it might be better to remove the name of each topic (i.e. night scene, summer activities, sitting and relaxation). If I understand LDA modelling correctly, it generates multiple clusters of topics automatically from a large number of texts while it can not interpret what that cluster is. The topic names (i.e. night scene, summer activities, sitting and relaxation) are likely given by the authors so that these names should better not be shown as part of the LDA diagram (figure 3).
- In line 161, are the “three landscape architecture researchers (inside)” the authors of this paper? If so, then say it directly: “the authors (three landscape architecture researchers)”.
To be considered:
- Sentences between line 279 to 284 seem to tell a lot of things while there is a lack of coherent and meaningful story; they are largely detached from this research, being so general that they might fit any papers on urban parks. I would suggest that some results of the LDA modelling and human interpretation (e.g. table 1) could be used here. For example, line 280 to 282 suggests that local businesses can learn from the findings of this paper but does not tell how (an example could be helpful here). In table 1, under topic 1 (holiday), ‘shop’, ‘market’, ‘stall’, ‘vendor’, and ‘food’ are identified as key terms. This indicates that a food market consisting of various temporary stalls is potentially a key attraction of Bryant Park in the holiday season (e.g. Christmas). This information could be useful for local retailers to organise their businesses in or around the park.
Author Response
I am largely satisfied with the revised article and think that the manuscript would be ready for publication once the following comments are followed and/or considered.
We really appreciate the reviewer’s time and effort to help improve our paper. We made all the changes the review suggested and we highlighted those changes with blue color in the newly submitted manuscript.
To be followed:
- The research gap is still murky, in particular the related sentence in the abstract (line 14): “Existing literature lacks comprehensive examinations of site qualities and experiences”; this is simply too broad. How about: “there is a lack of studies that investigate how people use and perceive urban parks using quantitative analysis of location-based social media reviews”? Accordingly, line 96 to 100 should be revised to clarify the research gap.
Thank you for the suggestion. We agree with the reviewer about line 14 and made the changes at as suggested. Regarding line 96- 100, we added more particular language and modified the content as “However, few studies have examined the perceived qualities and experiences using location-based social media reviews which provided essential information on public space usages and attachments. We tackle this gap by conducting a case study using online review data from TripAdvisor to investigate how users comment their experiences in Bryant Park, New York City”.
- In figure 3, it might be better to remove the name of each topic (i.e. night scene, summer activities, sitting and relaxation). If I understand LDA modelling correctly, it generates multiple clusters of topics automatically from a large number of texts while it can not interpret what that cluster is. The topic names (i.e. night scene, summer activities, sitting and relaxation) are likely given by the authors so that these names should better not be shown as part of the LDA diagram (figure 3).
The reviewer is correct that LDA only generate topics, and their interpretations are made by humans. Regarding figure 3, we agree that those topic names should be removed since they were made up by us. We’ve changed it accordingly.
- In line 161, are the “three landscape architecture researchers (inside)” the authors of this paper? If so, then say it directly: “the authors (three landscape architecture researchers)”.
Yes, they are the authors of this paper, we made the changes as suggested.
To be considered:
- Sentences between line 279 to 284 seem to tell a lot of things while there is a lack of coherent and meaningful story; they are largely detached from this research, being so general that they might fit any papers on urban parks. I would suggest that some results of the LDA modelling and human interpretation (e.g. table 1) could be used here. For example, line 280 to 282 suggests that local businesses can learn from the findings of this paper but does not tell how (an example could be helpful here). In table 1, under topic 1 (holiday), ‘shop’, ‘market’, ‘stall’, ‘vendor’, and ‘food’ are identified as key terms. This indicates that a food market consisting of various temporary stalls is potentially a key attraction of Bryant Park in the holiday season (e.g. Christmas). This information could be useful for local retailers to organise their businesses in or around the park.
Thank you for the comments. The sentences between line 279 and 284 were suggested by the previous reviewers who would like to see some general statements there. However, we do agree with your idea that to have some of specific LDA interpretations here. We like your suggestions to talk about T1, and we added this concept into Line 283 “For example, our T1(Holiday Favorite) which includes topic terms such as shop, market, stall, vendor, and food indicates that temporary vendors or food services could be great placemaking programs for local retailers to organize their businesses in or around the park.”